# Comparative analysis reveals distinctive epigenetic features of the human cerebellum

Elaine E. Guevara [1,2]*, William D. Hopkins[3], Patrick R. Hof[4,5,6], John J. Ely[7], Brenda J. Bradley[2], Chet C. Sherwood[2]

1 Department of Evolutionary Anthropology, Duke University, Durham, North Carolina, United States of America, 2 Department of Anthropology and Center for the Advanced Study of Human Paleobiology, The George Washington University, Washington, District of Columbia, United States of America, 3 Keeling Center for Comparative Medicine and Research, University of Texas MD Anderson Cancer Center, Bastrop, Texas, United States of America, 4 Nash Family Department of Neuroscience, Icahn School of Medicine at Mount Sinai, New York, New York, United States of America, 5 Friedman Brain Institute, Icahn School of Medicine at Mount Sinai, New York, New York, United States of America, 6 New York Consortium in Evolutionary Primatology, New York, New York, United States of America, 7 MAEBIOS, Alamogordo, New Mexico, United States of America

* eg197@duke.edu

**Data Availability Statement:** Metadata and raw spectral intensity data are available in NCBI's Gene Expression Omnibus (https://www.ncbi.nlm.nih.gov/sra) under accession GSE154403.

## Abstract

Identifying the molecular underpinnings of the neural specializations that underlie human cognitive and behavioral traits has long been of considerable interest. Much research on human-specific changes in gene expression and epigenetic marks has focused on the pre-frontal cortex, a brain structure distinguished by its role in executive functions. The cerebellum shows expansion in great apes and is gaining increasing attention for its role in motor skills and cognitive processing, including language. However, relatively few molecular studies of the cerebellum in a comparative evolutionary context have been conducted. Here, we identify human-specific methylation in the lateral cerebellum relative to the dorsolateral pre-frontal cortex, in a comparative study with chimpanzees (*Pan troglodytes*) and rhesus macaques (*Macaca mulatta*). Specifically, we profiled genome-wide methylation levels in the three species for each of the two brain structures and identified human-specific differentially methylated genomic regions unique to each structure. We further identified which differentially methylated regions (DMRs) overlap likely regulatory elements and determined whether associated genes show corresponding species differences in gene expression. We found greater human-specific methylation in the cerebellum than the dorsolateral prefrontal cortex, with differentially methylated regions overlapping genes involved in several conditions or processes relevant to human neurobiology, including synaptic plasticity, lipid metabolism, neuroinflammation and neurodegeneration, and neurodevelopment, including developmental disorders. Moreover, our results show some overlap with those of previous studies focused on the neocortex, indicating that such results may be common to multiple brain structures. These findings further our understanding of the cerebellum in human brain evolution.

**Funding:** The work was supported by funding from the Center for the Advanced Study of Human Paleobiology at The George Washington University (https://cashp.columbian.gwu.edu/) to CCS and EEG, Duke University Department of Evolutionary Anthropology to EEG (https://evolutionaryanthropology.duke.edu/), the James S. McDonnell Foundation (https://www.jsmf.org/) Grant #220020293 to CCS, and National Science Foundation (https://www.nsf.gov/) Grants SMA-1542848 to CCS, WDH, and BJB; EF-2021785 to CCS, and BSC-1919780 to CCS and EEG. The National Chimpanzee Brain Resource was supported by National Institutes of Health (https://www.nih.gov/) Grant NS092988 to CCS and WDH. The funders had no role in study design, data collection and analysis, decision to publish, or preparation of the manuscript.

**Competing interests:** The authors have declared that no competing interests exist.

## Author summary

Humans are distinguished from other species by several aspects of cognition. While much comparative evolutionary neuroscience has focused on the neocortex, increasing recognition of the cerebellum's role in cognition and motor processing has inspired considerable new research. Comparative molecular studies, however, generally continue to focus on the neocortex. We sought to characterize potential genetic regulatory traits distinguishing the human cerebellum by undertaking genome-wide epigenetic profiling of the lateral cerebellum, and compared this to the prefrontal cortex of humans, chimpanzees, and rhesus macaque monkeys. We found that humans showed greater differential CpG methylation–an epigenetic modification of DNA that can reflect past or present gene expression–in the cerebellum than the prefrontal cortex, highlighting the importance of this structure in human brain evolution. Humans also specifically show methylation differences at genes involved in neurodevelopment, neuroinflammation, synaptic plasticity, and lipid metabolism. These differences are relevant for understanding processes specific to humans, such as extensive plasticity, as well as pronounced and prevalent neurodegenerative conditions associated with aging.

## Introduction

Humans' remarkable cognitive abilities–enabling language use, complex technology, and cultural behavior—are hallmarks of our species. The neurobiological underpinning of these traits originate in large part from developmentally established species-specific spatial and temporal patterns of gene regulation in the brain [1–6]. One important gene regulatory mechanism critical to establishing species-typical and cell type-specific transcriptional profiles is CpG methylation. This epigenetic mark involves the addition of a methyl chemical group by methyltransferase enzymes to cytosine DNA bases occurring next to guanine bases, or CpG sites, and can affect gene expression by influencing transcription factor binding and chromatin organization [7]. Methylation across the genome can be considered to represent the footprint of developmentally configured, cell type-specific gene regulatory settings [8]. Consistent species differences in methylation may thus reflect evolved developmental differences.

As such, methylation, along with gene expression, has been studied in a comparative evolutionary context in an effort to identify clues to the molecular basis of human-specific traits. Indeed, differential methylation among closely related primate species has been identified in several tissues, including kidney, liver, blood, and bone [9–12]. Comparing brain methylation patterns among humans and other primates to reveal species differences in developmental programming and plasticity is thus of considerable interest in the search for what sets human cognition apart. Although nonhuman primates share the same brain structures as humans, some structures have taken on new roles that enable human-specific cognitive abilities and behaviors, which may be reflected in changes in relative size, microstructure, or connectivity [13]. For example, the question of whether humans have a relatively enlarged prefrontal cortex, a region that plays a role in language and abstract thought, has been the subject of much study, and the connectivity and neurotransmitter innervation of this structure appears to be distinct in humans [13,14].

Such changes in our lineage presumably have molecular bases and previous studies have found differential methylation between humans and chimpanzees in the prefrontal cortex associated with genes involved in developmental processes, psychiatric conditions, and neurodegeneration [15,16], as well as genes implicated in brain size [17] and regulation of the language-associated gene *FOXP2* [18]. Together, these findings affirm the potential relevance of methylation in brain tissue for understanding human neurobiology and evolution.

We build on this evidence by comparing genome-wide methylation in humans, chimpanzees (*Pan troglodytes*), and rhesus macaques (*Macaca mulatta*) in the dorsolateral prefrontal cortex (DLPFC), as well as the lateral cerebellum (Fig 1). The DLPFC is a specialized part of the prefrontal cortex involved in executive functions, which are important for working memory, attention, cognitive flexibility, and planning (for review, see [19]). It is also heavily interconnected with language areas of the cortex and striatum in humans [20]. Moreover, the human DLPFC is distinguished from other primates by patterns of cell type composition and distribution, as well as neurochemistry [21,22]. The DLPFC is also especially affected in autism spectrum disorder and schizophrenia [19]. The DLPFC has long been a focus of comparative evolutionary studies, including molecular analyses.

In contrast, the cerebellum has received comparatively little attention in human evolutionary neuroscience until fairly recently. Long recognized for its role in motor control, later discoveries of its role in non-motor function, including cognitive processes like language [23,24], spurred a surge of research on the cerebellum [25,26]. Intriguingly, comparative work has revealed that the lateral cerebellum is particularly enlarged in humans and other great apes [27–29], and its claim to the majority of the brain's neurons [30] has furthered interest in the cerebellum's role in human evolution. Indeed, it is increasingly recognized that the cerebellum likely plays a role in the fine motor skills necessary for speech and the precision and control underlying many cultural behaviors distinguishing humans [26]. Additionally, it is involved in cognitive processing of language [31,32] and executive function, with the cerebellum potentially allowing for a kind of supervised machine learning on spatiotemporal data input to build cognitive models and automate complex behaviors [25].

Nevertheless, comparative molecular studies have continued to focus predominately on the neocortex. In the current study, we thus sought to characterize epigenetic differences among humans and other species in the cerebellum to gain insights into potential human distinctiveness, as well as provide context to comparative studies of the neocortex. We specifically investigated the lateral cerebellum, given its enlargement in hominoids and extensive interconnection with the association areas of the neocortex [29].

Here, we examined species differences in methylation in the DLPFC and lateral cerebellum using a fairly large sample size of rare nonhuman primate samples and a large number ($> 160,000$) of CpG sites across the genome. Specifically, we identified human-specific differentially methylated genomic regions specific to each brain structure, employing a three-way comparison to identify whether differences between humans and chimpanzees occurred along the chimpanzee or human branch. We also determined whether species-specific patterns of methylation are unique to each brain structure or show overlap among structures or with blood, a non-neural tissue. We were especially interested in differentially methylated regions (DMRs) associated with genes and covering likely regulatory elements (e.g., near gene transcription start sites). We further determined whether genes associated with such regulatory-associated DMRs (regDMRs) showed species differences in gene expression.

## Results

### Differential methylation analysis

We analyzed DLPFC and cerebellum samples from seven human, eight chimpanzee, and seven rhesus macaque adults free of neuropathology (Methods; Fig 1A and S1 Table). We extracted DNA from 0.03–0.1 grams of tissue using the Qiagen DNeasy Blood and Tissue kit (Qiagen, Hilden, Germany), brought DNAs to a standard concentration (~70 ng/μl), and profiled genome-wide methylation levels from bisulfite-converted DNA by microarray (Illumina Infinium EPIC Methylation BeadChip; "EPIC array" hereafter). We retained only CpG sites

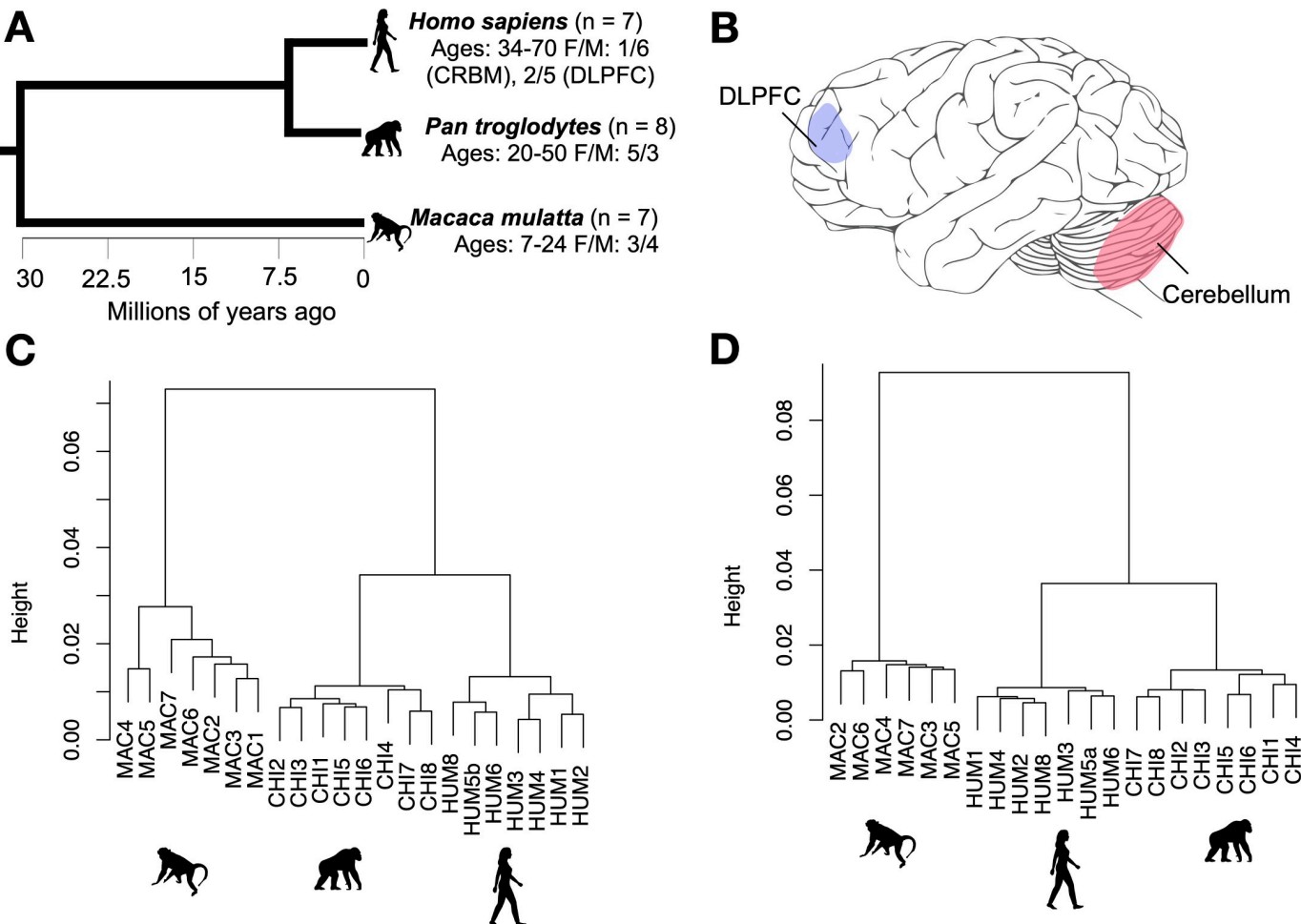

**Fig 1. Species relationships, brain structures, and sample clustering.** A = Phylogenetic tree of species included in this study from TimeTree.org [113], B = brain structures included in this study depicted on a chimpanzee brain illustration modified from [114], C and D = hierarchical clustering of samples from the dorsolateral prefrontal cortex (C) and cerebellum (D) based on genome-wide methylation after filtering and normalizing using correlations between samples as distance. DLPFC = dorsolateral prefrontal cortex.

expected to be efficiently captured across all three species in downstream analysis and used surrogate variable analysis to account for cell type heterogeneity (Methods). We identified human and chimpanzee-specific differentially methylated regions (DMRs) comprised of at least five CpGs, including at least three that showed an absolute difference in the proportion of methylation of at least 0.15 from the other two species. We identified several significant human- and chimpanzee-specific DMRs in each of the two brain structures (Table 1). Both species showed a greater number of DMRs in the cerebellum than in the DLPFC (human: DLPFC = 28, cerebellum = 64; chimpanzee: DLPFC = 5, cerebellum: 40). Many human- and brain structure-specific DMRs overlapped genes involved in processes and conditions of potential relevance to aspects of human-specific neurobiology, such as neurodevelopment, including neurodevelopmental disorders (Figs 2 and 3).

## Brain structure specificity and regulatory element association

Some (~7%) of the DMRs identified were evident in both brain structures (e.g., S1 Fig). We also compared our DMR dataset with a dataset of chimpanzee-human DMRs in blood. Of the

**Table 1. Human and chimpanzee brain region-specific regDMRs.**

| Brain structure | Species | Gene | Gene name | Hyper or hypo* |
|---|---|---|---|---|
| DLPFC | Human | PAQR4 KREMEN2 | progestin and adipoQ receptor family member 4 kringle containing transmembrane protein 2 | hyper |
| | | MYLK | myosin light chain kinase | hypo |
| | | ADAM30 | ADAM metallopeptidase domain 3 | mix |
| | | MLST8 | MTOR associated protein, LST8 homolog | hypo |
| | | TMEM177 | transmembrane protein 177 | hyper |
| | | TRIM65 | tripartite motif containing 65 | hyper |
| | | C10orf71 & C10orf71-AS1 | chromosome 10 open reading frame 71 | hyper |
| | | FAM193A | family with sequence similarity 193 member A | hypo |
| | | SLC13A4 | solute carrier family 13 member 4 | hyper |
| | Chimpanzee | LRCH1 | leucine rich repeats and calponin homology domain containing 1 | hyper |
| | | CUGBP1 | CUGBP Elav-like family member 1 | hyper |
| Cerebellum | Human | GP5 | glycoprotein V platelet | hyper |
| | | CHAD ACSF2 | chondroadherin acyl-CoA synthetase family member 2 | hypo |
| | | SEPP1 | selenoprotein P | hyper |
| | | DLEU7 | deleted in lymphocytic leukemia 7 | hyper |
| | | HRH1 | histamine receptor H1 | hyper |
| | | PARVG | parvin gamma | hypo |
| | | TRAF3IP2 | TRAF3 interacting protein 2 | hypo |
| | | SH3BGR | SH3 domain binding glutamate rich protein | hyper |
| | | FAM83A | family with sequence similarity 83 member A | hypo |
| | | ANKS1B | ankyrin repeat and sterile alpha motif domain containing 1B | hypo |
| | | DLGAP1 & DLGAP1-AS1/ FLJ35776 | DLG associated protein 1 | hypo |
| | | PLCH1 | phospholipase C eta 1 | hypo |
| | | FAM198B | golgi associated kinase 1B | hypo |
| | | TRAF3IP2 | TRAF3 interacting protein 2 | hypo |
| | | STK33 | serine/threonine kinase 33 | hypo |
| | | CCDC8 | coiled-coil domain containing 8 | hypo |
| | | GSTO2 | glutathione S-transferase omega 2 | hypo |
| | | CRIP3 | cysteine rich protein 3 | mix |
| | | CPNE6 | copine 6 | hypo |
| | | PHACTR1 | phosphatase and actin regulator 1 | hypo |
| | | FUT4 | fucosyltransferase 4 | hypo |
| | | MEDAG | mesenteric estrogen dependent adipogenesis | mix |
| | | MIR192 & MIR194-2 | microRNA 192 microRNA 194–2 | hypo |
| | | MIR1182 & FAM89A | microRNA 1182 family with sequence similarity 89 member A | hyper |
| | | DNAH10 | dynein axonemal heavy chain 10 | hyper |
| | | SDR42E1 | short chain dehydrogenase/reductase family 42E, member 1 | hyper |
| | | LOC101928322 | CELF2 divergent transcript | hypo |
| | | DMPK | DM1 protein kinase | hyper |
| | Chimpanzee | CCDC140 | CCDC140 long non-coding RNA | hyper |
| | | CCDC102A | coiled-coil domain containing 102A | hyper |
| | | LINGO3 | leucine rich repeat and Ig domain containing 3 | hypo |
| | | ZNF608 | Zinc finger protein 68 | hyper |
| | | MYO18A | myosin XVIIIA | hypo |

(*Continued*)

**Table 1.** (Continued)

| Brain structure | Species | Gene | Gene name | Hyper or hypo[*] |
|---|---|---|---|---|
| | | CDK15 | cyclin dependent kinase 15 | hypo |
| | | TRIM66 | tripartite motif containing 66 | hyper |
| | | SLC43A3 | solute carrier family 43 member 3 | hypo |
| | | ADAM32 | ADAM metallopeptidase domain 32 | hypo |
| | | SYTL1 | synaptotagmin like 1 | hyper |
| | | TCF21 | transcription factor 21 | hyper |
| | | BNC1 | basonuclin 1 | hyper |
| | | RNPEPL1 | arginyl aminopeptidase like 1 | hypo |
| | | APPL2 | adaptor protein, phosphotyrosine interacting with PH domain and leucine zipper 2 | hyper |
| | | GYPC | glycophorin C (Gerbich blood group) | hyper |
| | | DRD2 | dopamine receptor D2 | hypo |
| | | NODAL | nodal growth differentiation factor | hyper |
| | | ST3GAL1 | ST3 beta-galactoside alpha-2,3-sialyltransferase 1 | hyper |
| | | LRRC8D | leucine rich repeat containing 8 VRAC subunit D | hyper |
| | | TWIST2 | twist family bHLH transcription factor 2 | hyper |
| | | AVEN & CHRM5 | apoptosis and caspase activation inhibitor cholinergic receptor muscarinic 5 | hypo |
| | | LPL | lipoprotein lipase | hypo |
| | | KCNE3 | potassium voltage-gated channel subfamily E regulatory subunit 3 | hyper |

[*]Hypermethylated or hypomethylated

DLPFC DMRs, ~27% were also identified in blood, and, of the cerebellum DMRs, ~7% were also called as DMRs in blood. Five DMRs were shared across both brain structures and blood (e.g., S2 Fig). Overall, correspondence across tissues was high, with average global DLPFC and cerebellum methylation showing a correlation of 0.91–0.92 in all three species, DLPFC and blood methylation showing a correlation of 0.90–0.91 in humans and chimpanzees, and cerebellum and blood showing a correlation of ~0.82 in humans and chimpanzees, the two species represented in the blood dataset.

We next determined whether DMRs were associated with potential regulatory regions (Methods). In the DLPFC, nine human-specific and two chimpanzee-specific DMRs unique to the DLPFC were regulatory-associated DMRs (regDMRs; i.e., covering potential regulatory elements; see Methods; S2 Table and Figs 2 and 3). In the cerebellum, we identified 28 such regDMRs that were human-specific, which is over three times as many as in the DLPFC, and 23 that were chimpanzee-specific (Tables 1 and S2).

## Correspondence with gene expression and previous studies

We also identified whether genes associated with species-specific regDMRs showed species-specific differential gene expression using a previously published RNA-seq dataset including humans, chimpanzees, and rhesus macaques (Methods). We identified 281 genes that showed human-specific expression in the DLPFC and 487 that showed human-specific expression in the cerebellum. We also identified 231 genes that showed chimpanzee-specific expression in the DLPFC and 339 that showed chimpanzee-specific expression in the cerebellum. In the DLPFC, seven genes associated with human-specific, DLPFC-specific regDMRs were represented in the gene expression dataset. None showed differential gene expression in humans. Two genes associated with chimpanzee-specific, DLPFC-specific regDMRs were represented

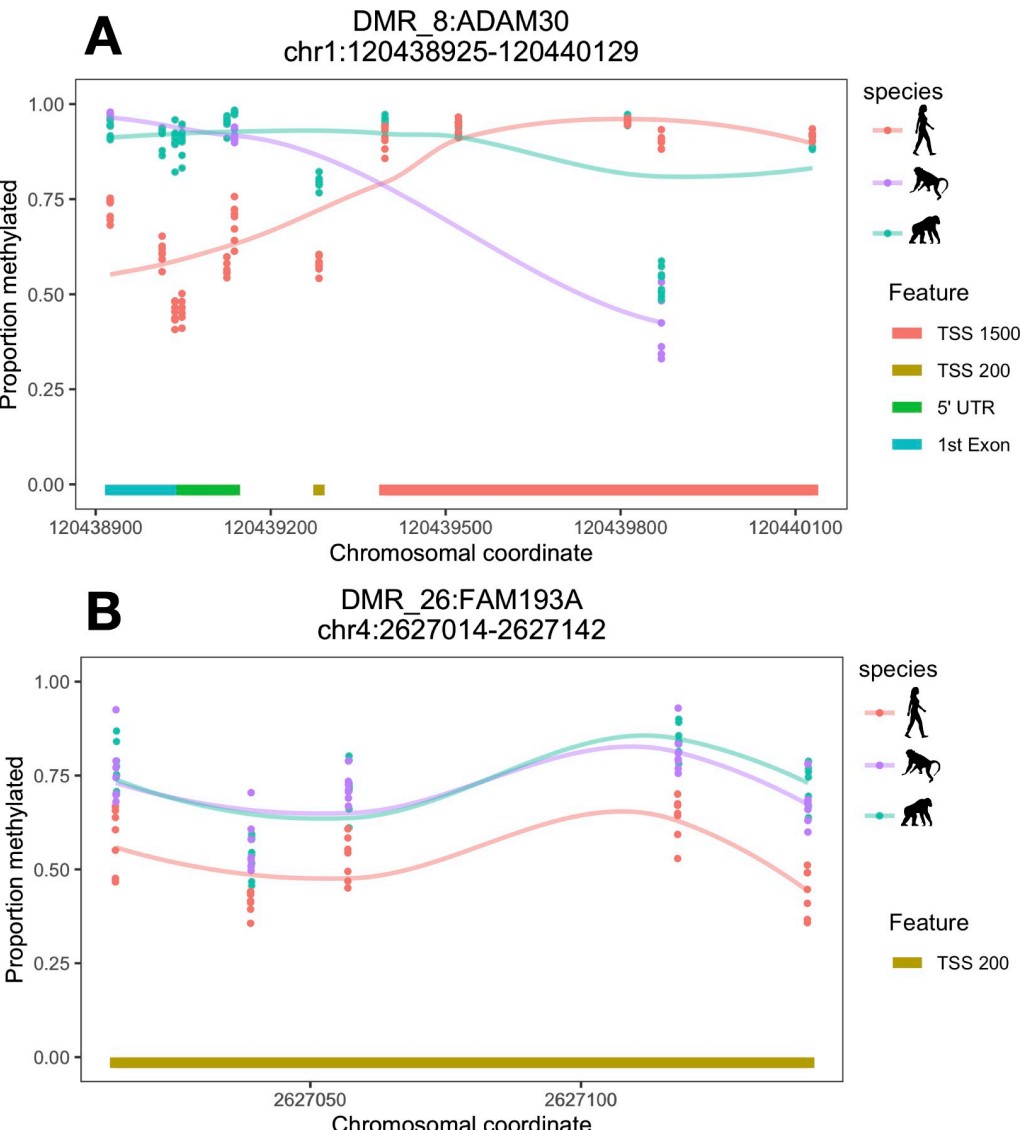

**Fig 2. DLPFC-specific human regDMRs associated with *ADAM30* and *FAM193A*.** A = *ADAM30*, B = *FAM193A*. Lines are loess smoothed methylation values for each species across the DMR and each point represents raw methylation values for each individual at each CpG site within the DMR ranges. TSS200 = within 200 bp of a transcription start site and TSS1500 within 1500 bp of a transcription start site. Chromosomal coordinates are in base pairs and refer to human genome build hg19.

in the gene expression dataset and neither was associated with differential gene expression. In contrast, in the cerebellum, five (*CHAD*, *STK33*, *CCDC8*, *CPNE6*, *SDR42E1*) of the 24 genes associated with human-specific, cerebellum-specific regDMRs that were represented in the gene expression dataset showed significant human-specific differential expression (Fig 4 and S2 Table), which represents more than would be expected by chance (Fisher's exact test; p = 0.004). Four of these genes show the canonical inverse relationship between methylation and gene expression, with three (*CHAD*, *STK33*, *CPNE6*) showing hypomethylation and upregulation (Fig 4) and one (*SDR42E1*) showing hypermethylation and downregulation. The fifth gene (*CCDC8*) unexpectedly shows the opposite pattern with hypermethylation and upregulation. Eighteen genes associated with chimpanzee-specific, cerebellum-specific regDMRs were

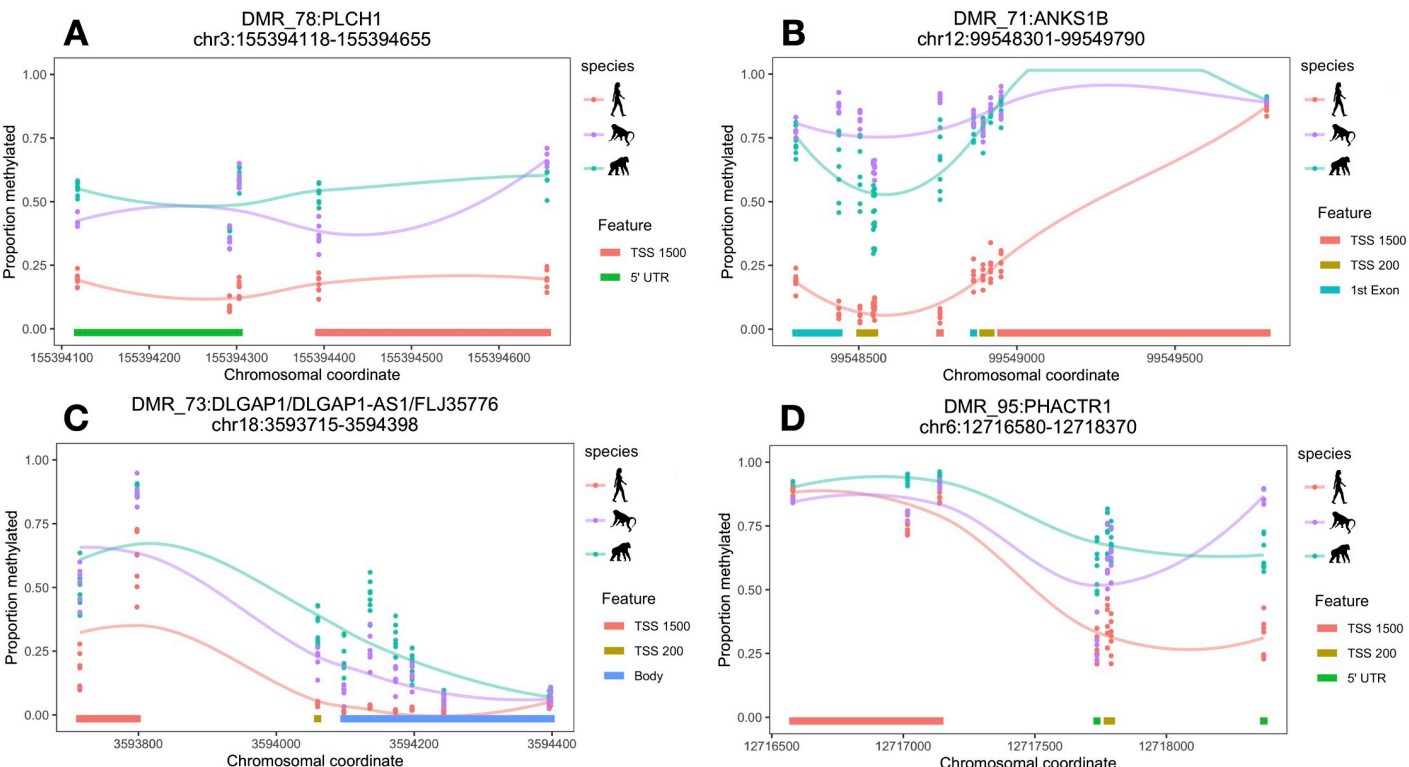

**Fig 3. Cerebellum-specific human regDMRs associated with *PLCH1*, *ANKS1B*, *DLGAP1*, and *PHACTR1*.** A = *PLCH1*, B = *ANKS1B*, C = *DLGAP1*, D = *PHACTR1*. Lines are loess smoothed methylation values for each species across the DMR and each point represents raw methylation values for each individual at each CpG site within the DMR ranges. TSS200 = within 200 bp of a transcription start site and TSS1500 within 1500 bp of a transcription start site. Chromosomal coordinates are in base pairs and refer to human genome build hg19.

represented in the gene expression data and two (*SYTL1* and *NODAL*) showed significant, chimpanzee-specific differential gene expression (S3 Fig and S2 Table). This degree of overlap is not more than what would be expected by chance (Fisher's exact test; p = 0.12). Both show inverse methylation and gene expression with *SYTL1* exhibiting hypomethylation and upregulation and *NODAL* hypermethylation and downregulation. Macaques, however, show no expression of *NODAL*, despite showing lower methylation than chimpanzees.

We also assessed whether genes showing uniquely human gene expression in the cerebellum shared the gene ontology (GO) term "chemical synaptic transmission" (GO:0007268) that annotated some of our cerebellar regDMR genes (S3 and S4 Tables). We found that 19 genes showing uniquely human gene expression are annotated with this GO term (S3 Table). However, this does not represent statistical enrichment (Fisher's exact test, p = 0.2763).

Finally, we assessed whether global methylation levels in our dataset showed correspondence with a previously published WGBS dataset for chimpanzees and humans. We found that average methylation levels across sites were highly correlated between our dataset and the WGBS dataset at overlapping CpGs (chimpanzees: Pearson's R = 0.91; humans: Pearson's R = 0.90).

## Discussion

Our results showing divergence in methylation of the cerebellum comparable to that in the DLPFC is notable giving a long-standing focus on the prefrontal cortex in comparative neuroscience. Moreover, we identified a greater number of human-specific regDMRs in the

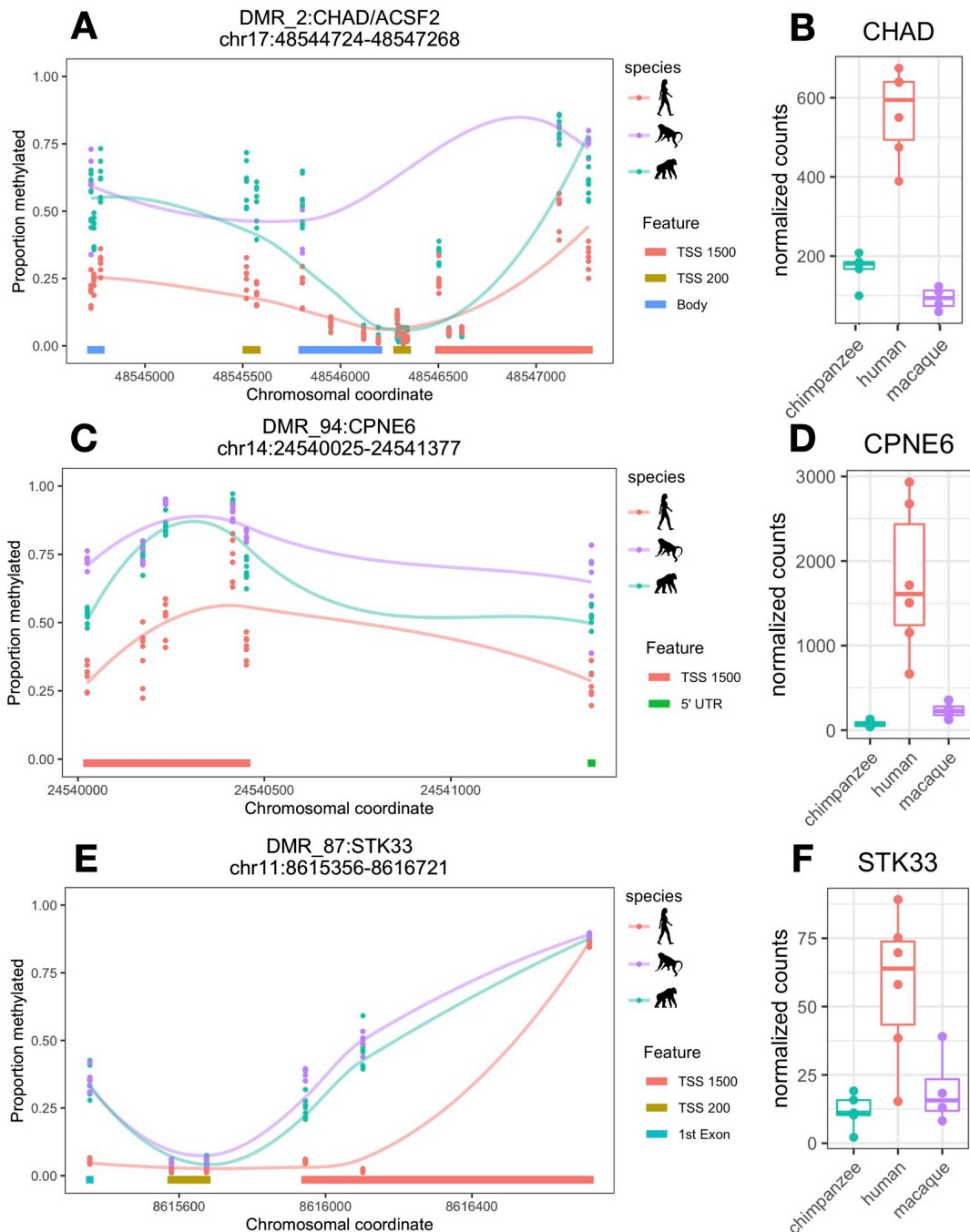

**Fig 4. Cerebellum-specific human regDMRs and gene expression levels for *CHAD*, *CPNE6*, and *STK33*.** A = *CHAD* methylation, B = *CHAD* gene expression, C = *CPNE6* methylation, D = *CPNE6* gene expression, E = *STK33* methylation, and F = *STK33* gene expression. Lines are loess smoothed methylation values for each species across the DMR and each point represents raw methylation values for each individual at each CpG site within the DMR ranges. TSS200 = within 200 bp of a transcription start site and TSS1500 within 1500 bp of a transcription start site. Chromosomal coordinates are in base pairs and refer to human genome build hg19.

cerebellum than the DLPFC, and some of these cerebellar human-specific regDMRs were associated with genes that also showed uniquely human gene expression in the cerebellum, which was not the case for the DLPFC. Our finding affirms the importance of the study of the

cerebellum in human brain evolution. This finding is consistent with recent analyses of gene expression across many brain structures [33] and the observed disproportionate expansion of the cerebellum in humans [27], as well as in line with increasing recognition of the cerebellum's potential importance in human-specific cognitive and behavioral traits, including language, speech production, and spatiotemporal processing [25–28,31,32].

Of the five human-specific differentially methylated regions associated with corresponding human-specific gene expression, several are not well characterized or do not have known roles in the brain. *CPNE6* (copine 6; Fig 4C and 4D), however, plays a role in synaptic plasticity and learning and, notably, was previously found to show prolonged developmental expression in the human cortex relative to nonhuman primates [34,35].

Several other human-specific regDMRs overlapped genes involved in neurodevelopment (*FAM198B*, *PHACTR1*, and *DIP2C* in the cerebellum) and neurodevelopmental disorders, including autism spectrum disorder (*ANKS1B*, *DIP2C*, and *DLGAP1* in the cerebellum) and schizophrenia (*FAM193A* in the DLPFC and *ANKS1B*, *DLGAP1*, and *DLEU7* in the cerebellum; Figs 2B and 3B and 3C) [36–43]. These results are consistent with previous studies [15,16] and may reflect species differences in development, as well as potentially greater human vulnerability to neuropsychiatric disorders [44–46].

Several genes overlapped by human-specific regDMRs are also implicated in neuroinflammation and neurodegeneration, including *ADAM30*, *GSTO2*, *PRSS50*, *MYOZ3*, and *ANKS1B* (Figs 2A and 3B) [47–51]. For example, one of the DLPFC regDMRs spans regulatory elements associated with *ADAM30* (ADAM metallopeptidase domain 30; Fig 2A), a gene that plays a role in processing amyloid precursor protein and appears to be downregulated in Alzheimer's disease [50]. A cerebellum-specific regDMR overlaps regulatory elements associated with *ANKS1B* (ankyrin repeat and sterile alpha motif domain containing 1B; Fig 3B), which is enriched at synapses in the cerebellum and binds amyloid precursor protein, decreasing amyloid beta protein production [38,48]. Humans may be specifically susceptible to dementia and neurodegeneration with age. Nonhuman primates show cognitive declines [52–55], age-related brain structural changes [56,57], and some degenerative pathology [58–60]. However, these phenotypes are generally considered to be mild compared to the severe degenerative pathology observed in humans [59,61,62]. Greater neurodegeneration in humans is thought to reflect a possible increase in oxidative stress and inflammation resulting from heightened energy metabolism over an extended lifespan [59,63,64].

Notably, a number of the genes associated with regDMRs (*CPNE6*, *DLGAP1*, *EGFLAM*, and *ANKS1B*) and implicated in neuropathology play roles in synaptic plasticity with several functioning as postsynaptic scaffold components [34,36,48,65]. Dysfunction in synaptic transmission and plasticity during development and adulthood may be a common risk in neurodevelopmental and degenerative disorders [48]. Greater and prolonged synaptic plasticity in humans has been hypothesized to underlie human behavioral and cognitive flexibility [64,66,67]. The cost of this increased, sophisticated cognitive dexterity may thus be another contributor to increased susceptibility to neurodevelopmental and degenerative conditions [66].

Lipid metabolism is also important in the brain as lipids play critical roles in the structure and function of nervous tissue. Notably, we identified a human-specific hypomethylated regDMR in the cerebellum overlapping *PLCH1* (Fig 3A), which encodes a phospholipase. *PLCH1* was previously found to show human-specific expression and to interact with lipids comprising part of the human-specific brain lipidome profile [68], although we did not identify a human-specific pattern of expression of this gene in the present analysis. Human-specific regDMRs were also associated with several noncoding RNA genes, including *LOC101928322*, *MIR192*, and *MIR1182*. The important roles of noncoding RNAs in the brain are increasingly gaining attention [69].

Chimpanzee-specific DMRs (Tables 1 and S2) were also of interest. Chimpanzees exhibit a cerebellum-specific DMR at *DRD2* (S4 Fig), the gene encoding the dopamine receptor D2, one of five receptors for the neurotransmitter dopamine [70]. The genetic modulation of dopaminergic pathways in intra- and interspecific variation in primate social behavior has garnered considerable interest [71–73]; however, the significance of such variation in methylation in the cerebellum requires further study.

Our data contained overall fewer CpGs than a comparative study of the prefrontal cortex in primates using whole genome bisulfite sequencing (WGBS) [15] such that none of the genomic ranges of DMRs identified in that study were represented in our data by enough CpGs (5) to comprise a DMR. We nevertheless found the genomic ranges of the previous study's human-specific DMRs to be enriched for individual, human-specific differentially methylated CpGs in our datasets for both brain structures (Fisher's exact test, $p < 0.001$), with slightly over half the CpGs present in each dataset falling within these ranges showing human-specific methylation. This finding, along with generally strong correlation of methylation levels between datasets across all overlapping CpGs, suggests that our study is broadly concordant with this previous study using a sequencing-based approach. Notably, however, our cerebellum dataset was as enriched for human-specific differentially methylated CpGs in these ranges as was the DLPFC dataset and two genes that we found to show human-specific methylation in the cerebellum (*DIP2C* and *RBPMS*) were previously associated with human-specific methylation in the prefrontal cortex [15]. These findings indicate that previously identified human-specific DMRs might not be unique to the prefrontal cortex, or even the neocortex.

Thus, our results suggest that human-specific DMRs previously identified in a particular brain structure may not actually be structure-specific and highlights the value of jointly comparing species and brain structure differences. This is consistent with our findings of substantial (~27%) overlap in human-specific differential methylation between the two brain structures, as well as less extensive, but still notable (~7%) overlap with differential methylation we identified in blood, suggesting that not all differential methylation in the brain necessarily reflects brain-specific regulatory differences. The correlations in global methylation between blood and the DLPFC and cerebellum are highly similar to those previously reported [74], as is the greater correspondence in methylation between blood and neocortex than blood and cerebellum [74,75] in global methylation, as well as DMRs. Overlapping differential methylation with blood might reflect human-specific patterns established early in development in cell types with a common developmental origin. Because there is blood present in the brain, some signal from blood methylation may also be detected in our brain data. Humans, furthermore, may show greater overlap among structures/tissues than chimpanzees because the array targets human functional regions.

Overlapping differential methylation across brain structures might reflect neuron-specific, species-specific patterns of methylation. Along these lines, some degree of divergence in differential methylation that we observed between brain structures might reflect differences in relative cell type composition. Although we have accounted for interindividual differences in cell type composition within regions using a surrogate variable approach, we have not accounted for cell type composition differences between regions. Notably, the cerebellum has a much higher proportion of neurons than the cerebral cortex and the composition of neurons relative to glia varies less among species with brain size [30]. Moreover, compared to the cerebral cortex, there is a higher relative proportion of inhibitory interneurons in the neocortex than the cerebellum [76]. As such, the cerebellum is generally more homogenous in cell composition within and across species, and observed species differences in the cerebellum may reflect neuron- or excitatory neuron-specific rather than region-specific differences. Single cell profiling would address this question.

The modest correspondence with gene expression in our analysis is consistent with the results of previous analyses. Prior studies of species differences in methylation have uncovered that methylation only explains a limited degree of variation in gene expression (e.g., 12–18%), even when gene expression and methylation levels are measured in the same tissue samples from the same individuals [11,15,77], which was not the case in our study. Rather, we leveraged a publicly available gene expression dataset matched for species and brain structure. It is not entirely surprising that gene expression at a given moment only tracks methylation to a limited degree given that methylation is only one of many regulatory mechanisms that influence gene expression and that variation in methylation and gene expression operate on different temporal scales, with methylation being relatively stable and gene expression being characterized by intermittent bursts of transcription, resulting in generally greater fluctuations in gene expression levels than methylation levels [11,78,79]. Furthermore, predicting whether methylation will lead to increased or decreased gene expression is complicated [79]. Although heavy methylation near transcription start sites canonically represses gene expression, hypomethylated regions, considered "transcriptionally permissive," can be associated with either transcriptional activity or inactivity [80,81]. In addition, promoter methylation has now been found to bind certain transcription factors [82], can also promote transcription when it occurs at insulator binding sites [83], and can sometimes be overridden by gene body methylation [79].

Moreover, methylation in adulthood may reflect vestiges of transcriptional states in development rather than or in addition to present transcriptional states [84,85]. In particular, previous studies have demonstrated that methylation in adulthood at developmentally expressed genes can provide a window into past gene expression in neurons [85], which is of interest in itself, given challenges and limitations to assaying methylation directly during development in humans and nonhuman primates, especially great apes. Thus, while we believe the species- and brain structure-specific regDMRs in our analysis that exhibit corresponding changes in gene expression are of great interest, we also think the regDMRs unassociated with differences in adult gene expression warrant consideration and are nevertheless promising candidates for further study, particularly given many of their associations with genes involved in neurodevelopment.

Our study has a number of important limitations. First, our use of a microarray for humans biases our results toward conserved and human functional regions. We are thus likely missing divergent regions, such as enhancers specific to the other species. This may be why we observed overall less chimpanzee-specific methylation and less overlap in chimpanzee-specific methylation between brain structures. Also, we did not assay non-CpG methylation. Methylation at cytosines outside of CpG contexts has an important role in the brain [86]; species comparisons in non-CpG methylation are therefore an important part of the picture that is absent from our current analysis. It is further impossible to distinguish presently between evolved differences in developmental programs and species differences that may be environmental in origin. Indeed, given the known role of methylation in environmental programming [87] and the protracted postnatal development of the brain [88], it is highly likely that growing up human in our highly constructed environments contributes to some of the species differences we observe, as it likely does to the unique cognitive traits we are interested in understanding. Both evolved and environmentally or culturally induced differences are of interest for understanding human cognitive uniqueness, but have different bases that are not readily teased apart. Finally, heterogeneity in sample collection and processing could influence some of our findings. Although the limited availability of human and nonhuman primate tissues is a challenge that can result in suboptimal experimental variation like that in our study [77], the relative stability of methylation, for example, relative to RNA, under various storage conditions, or its

"technical reliability" [79,89–92], makes it unlikely that batch effects resulting from differences in storage conditions are a major issue.

In conclusion, we identified human-specific patterns of methylation in the cerebellum, in addition to the DLPFC, which may reflect evolved and/or developmental neurobiological differences relevant to human-specific traits, like language, tool use, and cultural behavior, in these brain structures. Notably, we found greater human-specific differential methylation in the cerebellum. This finding may be driven by strong signal from excitatory neurons, which make up the bulk of cells in the cerebellum. Indeed, the cerebellum houses the majority of the brain's neurons and plays a role in many important cognitive, as well as motor, processes [25,26,31,32]. Our results lend further weight to growing interest in cerebellar evolution, as well as provide crucial comparative context for interpreting studies of differential methylation in the cortex. In addition, we found that human-specific differential methylation overlapped genes in pathways associated with neurodevelopment, which may reflect aspects of human-specific developmental trajectories, and synaptic plasticity and lipid metabolism, which are thought to be altered in humans [66,68]. These differences may hold clues to the molecular underpinnings of human cognitive specializations. Moreover, some of these genes are also implicated in neuropathology and thus may offer clues to human disease susceptibilities. Few genes showing human-specific methylation patterns at putative regulatory elements showed human-specific expression; nevertheless, these genes include some interesting candidates for further study, such as *CPNE6* (copine-6), which plays a role in learning and memory and shows human-specific hypomethylation and upregulation in the cerebellum. Our results highlight the value of tissue-specific species comparisons of methylation and are consistent with an important role for the cerebellum in human brain evolution.

## Methods

### Ethics statement

Brain specimens were obtained from the NIH NeuroBioBank, the National Chimpanzee Brain Resource, the Southwest National Primate Research Center, and the California National Primate Research Center with the approval of The George Washington University Institutional Animal Care and Use Committee (Protocol #A454). No living animals were used.

### Study subjects

Specimens were collected postmortem from 22 individuals of three species (7 humans, 8 chimpanzees, and 7 rhesus macaques) (Fig 1A and S1 Table) who died from causes unrelated to this research. All human specimens were categorized as healthy controls. Brain specimens were frozen after postmortem intervals (PMIs) of less than 24 hours, except in the case of one of the human individuals (PMI: 26.7 hours). We selected a mix of relatively young and older adult subjects from each species. We obtained the specimens from the NIH NeuroBioBank, the National Chimpanzee Brain Resource, the Southwest National Primate Research Center, and the California National Primate Research Center. Following necropsy, brain specimens were coronally sectioned and slabs were stored at -80˚C. Both brain regions came from the same individual, except in one case for humans, where the two regions came from different individuals of a similar age but different sex.

### Tissue dissection

The dorsolateral prefrontal cortex (DLPFC, corresponding to Brodmann's area 46) and lateral cerebellum (Crus I and Crus II; "cerebellum" hereafter) were dissected from frozen

chimpanzee and macaque brain sections kept chilled on dry ice using published species-specific brain atlases [27,93,94] (Fig 1B). Dissected tissue was then transferred to RNALater (Ambion, Austin, TX, USA) preservation buffer and stored at -20˚C in solution until DNA extraction. Human samples from the structures of interest were requested from the NIH NeuroBioBank and anatomical dissections followed by snap-freezing and pulverization were carried out by their staff. All tissue derived from the left hemisphere, except for in the case of two of the chimpanzees for which only the right hemisphere was available.

## DNA extraction and microarray analysis

DNA was extracted using the DNeasy Blood and Tissue kit (Qiagen, Hilden, Germany). Prior to extraction, RNALater was washed from macaque and chimpanzee samples with PBS buffer. We quantified DNA extracts on a Nanodrop spectrophotometer (Thermo Fisher Scientific, Waltham, MA, USA) and brought them to a concentration of ~70 ng/µl, either through dilution with PCR-grade water or concentration using Microcon-30kDa Centrifugal Filter Unit columns (Millipore Sigma, Burlington, MA, USA). DNA was then bisulfite-converted and genome-wide methylation levels were profiled by microarray (Illumina Infinium EPIC Methylation BeadChip; "EPIC array" hereafter) at the Yale Center for Genome Analysis. A mix of species, ages, and sexes were profiled on each chip to avoid batch effects.

## Data preprocessing

We first filtered the raw intensity data to remove probes with spectral intensities not significantly different from background levels, that do not target CpG dinucleotides, that contain known SNPs, that are on the sex chromosomes, and for which fewer than 3 beads were counted for 5% or more of the samples with the ChAMP v2.18.3 [95] R package. We excluded one macaque cerebellum sample due to a high proportion of probes that did not show spectral intensities above background levels. We then normalized the data using Beta Mixture Quantile dilation (BMIQ), which accounts for the two different probe types on the EPIC array [96]. Because the Illumina methylation microarrays are designed to target human sites, we limited all analyses to CpG sites that should also be efficiently targeted in chimpanzees and rhesus macaques. Specifically, we restricted the dataset to CpG sites targeted by probes that map to the chimpanzee (panTro2.1.4) and rhesus macaque (Mmul8.0.1) genomes with 0–2 mismatch [s], without any mismatches within 5 bp of the target CpG, and without known SNPs in the target species following [12].

Brain tissue is a composite of multiple cell types. We used surrogate variable analysis, a robust, reference-free approach that can account for cell type heterogeneity [80,97]. Specifically, for both our DLPFC and cerebellum data separately, we first estimated the number of cell types (k) in the data using a random matrix theory approach to identify the number of variance components in the data above the number expected under a Gaussian Orthogonal Ensemble (GOE) using the EstDimRMT function in R package isva v1.9 [98]. EstDimRMT returned an optimal k of two variance components/cell types for each tissue. We then estimated two surrogate variables using a re-weighted least squares approach [99] with the sva function in the R package sva v3.36.0 [100]. We used singular value decomposition to assess remaining batch effects [101] in both the DLPFC and cerebellum datasets with the champ.SVD function in the ChAMP package v2.18.3 [95] and found no significant variance associated with array or slide.

## Differential methylation analysis

We identified differential methylation in two ways. We first identified differential methylation between humans and the two nonhuman primates, chimpanzees and macaques, in a dataset of

CpGs present in all three species. These datasets consisted of 165,398 and 165,740 probes for the DLPFC and cerebellum, respectively. The sparsity of CpGs in the dataset including macaques due to decreased sequence conservation with phylogenetic distance, however, limits our power to discover stretches of differential methylation in all three species. We thus additionally took a second, two-step approach to identifying species-specific methylation patterns in which we identified differential methylation between humans and chimpanzees only. After filtering, 556,428 and 557,796 probes remained in our chimpanzee-human DLPFC and cerebellum datasets, respectively. We then used overlapping, sparser macaque data to polarize these differences.

In both approaches, we first identified CpG sites showing differential methylation (differentially methylated positions, or DMPs) among species in each brain structure using the lmFit linear regression function in the limma v3.44.3 R package with eBayes coefficient shrinkage [102]. We included the covariates of age class, sex, and hemisphere, as well as the two surrogate variables to account for cell type heterogeneity (S1 Table). We considered DMPs significant at a false discovery rate [103] of 5%. We further required that DMPs exhibit a mean beta difference between humans and nonhuman primates of at least 0.15.

We next identified differentially methylated regions (DMRs)—or contiguous stretches of the genome enriched for differential methylation—among species using the DMRcate v2.2.3 R package, which calls DMRs based on Gaussian kernal smoothed methylation levels across stretches of the genome [104]. We required DMRs to contain at least five CpGs and that at least three of these CpGs be DMPs, as determined in our previous analysis. For the human-chimpanzee DMRs, we then determined whether human-chimpanzee differential methylation was human- or chimpanzee-specific by polarizing our comparisons using the macaque data. Specifically, we determined which human-chimpanzee DMRs overlapped at least three macaque-human or macaque-chimpanzee DMPs identified following the procedure described above and whether these DMPs represented at least 50% of the CpGs comprising the DMR or covered at least 50% of the DMR's length. The resulting DMRs were determined to be human- or chimpanzee-specific DMRs.

## Brain structure specificity and regulatory element association

We determined the degree to which DMRs in each structure were specific to that structure by assessing the overlap between the two sets of brain structure DMRs, as well as their correspondence with DMRs in blood, using a human and chimpanzee dataset. Blood methylation data for chimpanzees comprised a dataset we previously generated for 87 chimpanzee blood samples from 73 individuals using the EPIC array [105]. Human blood methylation data generated on the 450K array was downloaded for 274 individuals [106–108] from the Gene Expression Omnibus online database (accessions: GSE40279, GSE87571, GSE56105). Only probes on both the EPIC and 450K arrays that are expected to effectively target CpG sites also present in our brain methylation dataset were retained in this analysis. The final dataset was comprised of 129,003 probes. Data were quality filtered and normalized, surrogate variables estimated to account for cell type heterogeneity, and DMPs and DMRs identified using the same procedure as for the brain data. We then determined overlap between blood and brain DMRs, as well as between the two brain structures. DMRs were considered to be overlapping if they shared more than half of their constituent CpGs or half of their genomic range with a DMR in another tissue. DMRs were considered to be structure- or tissue-specific if they did not show overlap with a DMR in the other brain structure or blood.

We further determined which genomic elements DMRs spanned and which genes, if any, they were associated with by annotating CpGs within DMRs using the Illumina EPIC array

annotation in the ChAMP v2.18.3 R package [95]. We considered DMRs comprised by DNA within 1500 bp of transcription start sites or covering untranslated regions or the first exon to be potentially regulatory associated (regDMRs). DMRs covering gene bodies or in intergenic regions were not considered to be regDMRs.

## Correspondence with gene expression

We assessed correspondence with gene expression of human and chimpanzee brain structure-specific regDMRs using previously published data for the two structures for the three species [109] (http://evolution.psychencode.org/#). We downloaded count data and analyzed it using the DESeq2 v1.22.2 R package [110]. We removed genes with fewer than ten counts and used the DESeq function to correct for differences in count due to library size, estimate data dispersion, and fit a differential expression model for each pair of species. We then identified differentially expressed genes in humans as genes that were significantly differentially expressed (adjusted p-value < 0.05) and exhibited a log fold change of at least 1.5 in both the human-chimpanzee and human-macaque comparison. We also identified genes that were differentially expressed in chimpanzees using an equivalent procedure. We also examined whether genes that show uniquely human expression patterns and are associated with human-specific regDMRs fall in the same pathways. Specifically, we determined whether any genes annotated with the biological process GO term "chemical synaptic transmission" (GO:0007268)–a term that annotated two genes (*DLGAP1*, *CPNE6*) associated with human-specific regDMRs in the cerebellum–showed uniquely human expression patterns in the cerebellum. We retrieved a list of genes annotated with this term using the get_anno_genes function in the R package GOfuncR v1.8.0 [111] and assessed whether these genes showed a uniquely human pattern of expression.

## Correspondence with previous studies

We assessed the overall correspondence of our DLPFC data with previously published prefrontal cortex WGBS data for humans and chimpanzees [16]. We calculated the mean methylation level for each species at each CpG site in each dataset. The chimpanzee WGBS data was aligned to the panTro2.0 genome build, so we used the liftover function in the R package rtracklayer v1.48.0 [112] to identify the corresponding hg19 coordinates. We then assessed the correlation between average methylation for each species in the two datasets across all sites overlapping between the two datasets. We used to same approach to measure correspondence of the prefrontal cortex WGBS data with our cerebellum data to assess region specificity.

## Supporting information

**S1 Fig. Human-specific regDMRs in both brain structures.** A = *PRSS50* and B = *EGFLAM*. Lines are loess smoothed methylation values for each species across the DMR and each point represents raw methylation values for each individual at each CpG site within the DMR ranges. TSS200 = within 200 bp of a transcription start site and TSS1500 within 1500 bp of a transcription start site. Chromosomal coordinates are in base pairs and refer to human genome build hg19.
(TIF)

**S2 Fig. Human-specific cross-tissue regDMRs.** A = *C1orf65* and B = *BMPER*. Lines are loess smoothed methylation values for each species across the DMR and each point represents raw methylation values for each individual at each CpG site within the DMR ranges. TSS200 = within 200 bp of a transcription start site and TSS1500 within 1500 bp of a

transcription start site. Chromosomal coordinates are in base pairs and refer to human genome build hg19.
(TIF)

**S3 Fig. Cerebellum-specific chimpanzee regDMRs associated with *STYL1* and *STYL1* gene expression.** A = methylation, B = gene expression. Lines are loess smoothed methylation values for each species across the DMR and each point represents raw methylation values for each individual at each CpG site within the DMR ranges. TSS200 = within 200 bp of a transcription start site and TSS1500 within 1500 bp of a transcription start site. Chromosomal coordinates are in base pairs and refer to human genome build hg19.
(TIF)

**S4 Fig. Chimpanzee-specific DMR overlapping *DRD2* in the cerebellum.** Lines are loess smoothed methylation values for each species across the DMR and each point represents raw methylation values for each individual at each CpG site within the DMR ranges. Chromosomal coordinates are in base pairs and refer to human genome build hg19.
(TIF)

**S1 Table. Study subjects.**
(XLSX)

**S2 Table. Species and region-specific DMRs.**
(XLSX)

**S3 Table. Genes showing uniquely human expression in the cerebellum annotated with the GO term chemical synaptic transmission.**
(XLSX)

**S4 Table. Human and chimpanzee brain region-specific regDMRs annotated with GO terms.**
(XLSX)

## Acknowledgments

We thank Dr. Soojin Yi for providing processed WGBS data; Cheryl Stimpson for expert technical assistance; NIH NeuroBioBank for providing human tissue samples; National Chimpanzee Brain Resource, Southwest National Primate Research Center and California National Primate Research Center for the rhesus macaque tissue samples; and Dr. Mary Ann Raghanti for helpful discussion.

## Author Contributions

**Conceptualization:** Elaine E. Guevara, Chet C. Sherwood.

**Formal analysis:** Elaine E. Guevara.

**Funding acquisition:** Elaine E. Guevara, William D. Hopkins, Brenda J. Bradley, Chet C. Sherwood.

**Investigation:** Elaine E. Guevara, Chet C. Sherwood.

**Resources:** William D. Hopkins, Patrick R. Hof, John J. Ely, Chet C. Sherwood.

**Supervision:** William D. Hopkins, Brenda J. Bradley, Chet C. Sherwood.

**Visualization:** Elaine E. Guevara.

**Writing – original draft:** Elaine E. Guevara, Patrick R. Hof, Chet C. Sherwood.

**Writing – review & editing:** Elaine E. Guevara, William D. Hopkins, Patrick R. Hof, John J. Ely, Brenda J. Bradley, Chet C. Sherwood.

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
