## [Decision Letter · Decision Letter 0]

11 Sep 2020

Dear Dr Guevara,

Thank you very much for submitting your Research Article entitled 'Comparison with chimpanzees and rhesus macaques reveals distinctive epigenetic features of the human prefrontal cortex and cerebellum' to PLOS Genetics. Your manuscript was fully evaluated at the editorial level and by independent peer reviewers. The reviewers appreciated the attention to an important problem, but raised some substantial concerns about the current manuscript. Based on the reviews, we will not be able to accept this version of the manuscript, but we would be willing to review again a much-revised version. We cannot, of course, promise publication at that time.

If you decide to revise the manuscript for further consideration at PLOS Genetics, please aim to resubmit within the next 60 days, unless it will take extra time to address the concerns of the reviewers, in which case we would appreciate an expected resubmission date by email to plosgenetics@plos.org.

[LINK]

We are sorry that we cannot be more positive about your manuscript at this stage. Please do not hesitate to contact us if you have any concerns or questions.

Yours sincerely,

Takashi Gojobori

Associate Editor

PLOS Genetics

Bret Payseur

Section Editor: Evolution

PLOS Genetics

Editor's comments:

In this paper, the authors studied epigenetics related to the evolution of human neural specialization, measuring genome-wide DNA methylation levels in brain tissues from humans, chimpanzees, and macaques. Their findings of evolutionary changes in species-specific methylation with its implication to neurobiological function are significant and interesting.  All the three reviewers agreed with this point.

However, Reviewers 2 and 3 questioned the authors’ usage of the “EPIC microarray“ rather than whole-genome bisulfite sequencing for a less biased data set.  In particular, Reviewer 2 suggested rejection of this manuscript, pointing out that the choice to use EPIC array to investigate the brain epigenomic and the absence of any cell (nuclei) type enrichment significantly reduced the potential impact of this study.  Reviewer 3 also stated that the major methodological issue that one see with this study is profiling methylation sites by microarray rather than whole-genome bisulfite sequencing for a less biased data set.

Therefore, I would like to see the authors’ response to the above-mentioned points regarding the methodology.

Moreover, I have to agree with the following particular points of Reviewers 1 and 3:

As Reviewer 1 pointed out that the statistical analyses employed need to be more detailed, and potential confounding batch effects should be further considered. This Reviewer further continued that additionally, as the authors note, research on DNA methylation in brain tissues from primates has been previously examined, so the originality of this study should be more explicitly stated and reinforced.

Reviewer 3 also stated that a more focused approach would have served this study better: Identify the areas of greatest difference between species, narrow down on the most interesting or understudied pathways, and experimentally test whether hypo- or hypermethylation correlates with increased or decreased expression of the genes in question to construct narrow hypotheses around these specific data.

Those comments should be taken into account seriously by the authors when they make revision of the manuscript.

In summary, I would request the authors to make major revision and send it back to us.  Then, the revised manuscript will be examined by the reviewers again, for making final decision.

Reviewer's Responses to Questions

**Comments to the Authors:**

Reviewer #1: Review is attached.

Reviewer #2: The study by Guevara et al, compared DNA methylation pattern for two brain regions: DLPFC, cerebellum across human, chimpanzees, and an old-world monkey rhesus macaques. This was a well-designed study to investigate the evolution of the epigenome and gene regulation in primates. The study included an impressive set of primary brain tissue specimens, with sample quality control and dissection carefully performed. In addition, DLPFC and cerebellum samples were collected from paired individuals, increasing the consistency of the dataset, and potentially allow the controlling of individual sample difference with regression models. However, the choice to use EPIC array to investigate the brain epigenomic and the absence of any cell (nuclei) type enrichment significantly reduced the potential impact of this study.

1. A number of groups (e.g. Guo 2013 Nat Neuro, Lister 2013 Science, Gabel 2015 Nature) have reported the presence of a large quantity of non-CG methylation in mammalian brains, which appears to be a very unique characteristic of the brain compared to other tissues. Likely non-CG methylation is relevant to the evolution of the brain given its uniquely high level of accumulation in the brain. Given these findings, the authors should consider using whole-genome bisulfite sequencing that can more effectively measure non-CG methylation. If cost is a concern here, what about EPIC capture sequencing assay?

2. It is well known that brain cell types have very diverse methylation profiles, especially between neuronal cells and glial cells. It is fully feasible to separate neuronal and non-neuronal cells using anti-NeuN antibody. Kozlenkov et al., 2015 NAR has also developed a method to separate excitatory and inhibitory neuronal nuclei using antibodies against transcriptional factor proteins. These methods have been tested with primary human brain tissue and should be readily applicable to primate brain tissues.

3. Separating neurons and glia would be important not just for extracting cell type specific signature, but also to ensure the inter-species difference was not due to the variable ratio between neuronal and non-neuronal cells. Differences in cell-type composition is a major confounding factor for epigenetic epidemiology studies and may have an impact on inter-species comparison too.

4. The manuscript should include some technical description of the specimens, the method for methylation profiling and a brief discussion of the statistical method for DMR calling at the beginning of the result section. It was difficult to put the scale of DMR numbers in the context with no technical description at all in the main text.

Reviewer #3: The manuscript by Guevara et al. presents a relevant contribution to species-and tissue-related differences in global methylation of the dorsolateral prefrontal cortex and lateral cerebellum, brain areas considered of specific importance in the evolution of human cognition. Identified differentially methylated regions included genes that encode factors relevant to energy metabolism, neuroinflammation, neural degeneration, and neurodevelopment. Diminishing the impact of these discoveries, human-specific differentially methylated regions showed extensive overlap with those identified in blood, suggesting that some epigenetic marks may be common to nonneural cell types. Further diminishing the impact of this work, only modest correspondence between human-specific differential methylation and differential gene expression was observed. While the overall scope of work appears appropriate and some new insight may have been gained here, overall advancement for the field appears modest and somewhat dampens enthusiasm for this paper. It appears to me that a more focused approach would have served this study better: Identify the areas of greatest difference between species, narrow down on the most interesting or understudied pathways, and experimentally test whether hypo- or hypermethylation correlates with increased or decreased expression of the genes in question to construct narrow hypotheses around these specific data. Please see below a more detailed critique.

The major methodological issue that I see with this study is profiling methylation sites by microarray rather than whole-genome bisulfite sequencing for a less biased data set. The authors explain the problem themselves “Because the Illumina methylation microarrays are designed to target human sites, we limited all analyses to CpG sites that should also be efficiently targeted in chimpanzees and rhesus macaques.” The bias that this approach introduces cannot possibly overstated and may stood in the way of discovering overlooked aspects of human-specific methylation and gene regulation.

The experimental design remains somewhat obscure and I recommend the authors provide additional clarifications. I assume that the numbers of DMPs and DMRs provided inn Fig. 1 and Table 1 are in relation to the reference methylome of macaque, but nowhere is this clearly explained. Please clarify.

The authors discuss multiple very interesting pathways involving genes that associate with DMRs, such as glucose metabolism, neuroinflammation etc., but do not provide data on whether the genes in question are upregulated or downregulated in these cases, which seems to me the most relevant question.

I recommend that whenever in the results the number of overlapping DMRs is provided, to also indicate what percentage of the total relevant DMRs this number corresponds to.

Results of figures 3 and 4 are only mentioned in the discussion and there only partially. Please present these data in the results section as well.

In the discussion the authors state: “Many human-specific DMRs also overlapped genes involved in neurodevelopment and neurodevelopmental disorders, especially autism spectrum disorder (ANKS1B, GCNT2, DIP2C, RAI1, MYOZ3, DIP2C, DLGAP1, ZNF8) and schizophrenia (NT5DC2, SNED1, RAI1, MYOZ3, FAM193A).” How do the listed genes associate with the ID and ASD modules established by Parikshak et al. 2013? Are prenatal or postnatal modules of greater relevance?

**Have all data underlying the figures and results presented in the manuscript been provided?**

Reviewer #1: Yes

Reviewer #2: Yes

Reviewer #3: Yes

PLOS authors have the option to publish the peer review history of their article (what does this mean?). If published, this will include your full peer review and any attached files.

Reviewer #1: No

Reviewer #2: No

Reviewer #3: No

---

## [Decision Letter · Decision Letter 1]

17 Feb 2021

Dear Dr Guevara,

Thank you very much for submitting your Research Article entitled 'Comparative analysis reveals distinctive epigenetic features of the human cerebellum' to PLOS Genetics.

The manuscript was fully evaluated at the editorial level and by independent peer reviewers. The reviewers appreciated the attention to an important topic but identified some concerns that we ask you address in a revised manuscript

We therefore ask you to modify the manuscript according to the review recommendations. Your revisions should address the specific points made by each reviewer.

[LINK]

Yours sincerely,

Takashi Gojobori

Associate Editor

PLOS Genetics

Bret Payseur

Section Editor: Evolution

PLOS Genetics

Reviewer's Responses to Questions

**Comments to the Authors:**

Reviewer #1: The authors have satisfactorily addressed all of my prior comments. In particular, the refocusing of the paper to emphasize the cerebellum very much clarifies and enhances the originality of this study. The additional details provided in the methods are also an excellent improvement. Finally, while the authors do a nice job of highlighting their results and the insights gained from them, they are also clear about the limitations of this work.

Minor Comments

[line 472] What does “age class” mean? If this is something different than age in years, please specify this variable in Table S1.

Some minor typos are present throughout the paper and should be corrected. For example:

• [line 108] “languages areas” should be changed to “language areas”

• [line 113] remove the comma between “molecular” and “studies”

• [line 119] add a comma after in-text citation “[27-29]”

Reviewer #3: The substantially revised manuscript by Guevara et al. presents an partial improvement over the initial submission addressing some points raised by the reviewers, particularly regarding methodological detail. However, some major points were not addressed, including my earlier suggestion to provide a greater focus to this study by identifying the areas of greatest difference between species, narrow down on the most interesting or understudied pathways, and experimentally test whether hypo- or hypermethylation correlates with increased or decreased expression of the genes in question to construct narrow hypothesis around these specific data. While I am very sympathetic to the argument about current limitations placed on most of us, even the existing data without additional experiments may have sufficed to meet part of this suggestion. For instance, I find Table 1 dissatisfying by not listing genes according to relevant gene ontology domains using common tools that may further highlight pathways of relevance.

In addition, the paper’s focus has been shifted by emphasizing the cerebellum as a structure of potential significance to human evolution, but throughout the manuscript and particularly so in the discussion this position is continuously weakened or the evidence supporting a disproportionate significance of the human cerebellum not sufficiently highlighted.

The results section has been shortened to a degree that makes it difficult to follow the process and rationale guiding experiments. Providing some additional detail, currently dispersed in introduction and methods should help. For instance, the results section is initiated by directly presenting the methylation data with no mention of the process leading to these data. While the introduction and methods sections provide more details, some of it has to make it into the results section as well. Specimen selection, amount of tissue collected, lysis procedure, and reference to the chip used for analysis are some essentials to provide context. In the process, please also include a reference to Fig 1 in the results section.

Line 165: Many (~7%) of DMRs identified were evident in both brain structures. I don’t think that 7% qualifies as many. I think “some” or a “minority” seems more appropriate.

Line 169: Overall, correspondence across tissues was high, with average global DLPFC and cerebellum methylation showing a correlation of 0.91-0.92 in all three species.

Line 208: Substantial species divergence in methylation of the cerebellum is notable giving a long standing focus on the prefrontal cortex in comparative neuroscience. Our finding affirms the importance of the cerebellum in human brain evolution.

The above two statements from results and discussion respectively disagree in my opinion. At .91-.92 correlation of both DLPFC and cerebellum between species, both structures show comparable divergence. Also, I am not sure if this level of between species divergence can be considered substantial, as it corresponds well to that seen in blood too.

**Have all data underlying the figures and results presented in the manuscript been provided?**

Reviewer #1: Yes

Reviewer #3: Yes

PLOS authors have the option to publish the peer review history of their article (what does this mean?). If published, this will include your full peer review and any attached files.

Reviewer #1: No

Reviewer #3: No

---

## [Editor Report · Decision Letter 2]

24 Mar 2021

Dear Dr Guevara,

We are pleased to inform you that your manuscript entitled "Comparative analysis reveals distinctive epigenetic features of the human cerebellum" has been editorially accepted for publication in PLOS Genetics. Congratulations!

Yours sincerely,

Takashi Gojobori

Associate Editor

PLOS Genetics

Bret Payseur

Section Editor: Evolution

PLOS Genetics

Comments from the reviewers (if applicable):

**Data Deposition**

http://datadryad.org/submit?journalID=pgenetics&manu=PGENETICS-D-20-01121R2

**Press Queries**

---

## [Editor Report · Acceptance letter]

13 Apr 2021

PGENETICS-D-20-01121R2 

Comparative analysis reveals distinctive epigenetic features of the human cerebellum 

Dear Dr Guevara, 

We are pleased to inform you that your manuscript entitled "Comparative analysis reveals distinctive epigenetic features of the human cerebellum" has been formally accepted for publication in PLOS Genetics! Your manuscript is now with our production department and you will be notified of the publication date in due course.

With kind regards,

Katalin Szabo

PLOS Genetics

On behalf of:
